# Role of Virally-Encoded Deubiquitinating Enzymes in Regulation of the Virus Life Cycle

**DOI:** 10.3390/ijms22094438

**Published:** 2021-04-23

**Authors:** Jessica Proulx, Kathleen Borgmann, In-Woo Park

**Affiliations:** 1Department of Pharmacology and Neuroscience, University of North Texas Health Science Center, Fort Worth, TX 76107, USA; jessicaproulx@my.unthsc.edu (J.P.); kathleen.borgmann@unthsc.edu (K.B.); 2Department of Microbiology, Immunology and Genetics, University of North Texas Health Science Center, Fort Worth, TX 76107, USA

**Keywords:** viruses, ubiquitination, ubiquitin proteasome system, deubiquitinases (DUBs)

## Abstract

The ubiquitin (Ub) proteasome system (UPS) plays a pivotal role in regulation of numerous cellular processes, including innate and adaptive immune responses that are essential for restriction of the virus life cycle in the infected cells. Deubiquitination by the deubiquitinating enzyme, deubiquitinase (DUB), is a reversible molecular process to remove Ub or Ub chains from the target proteins. Deubiquitination is an integral strategy within the UPS in regulating survival and proliferation of the infecting virus and the virus-invaded cells. Many viruses in the infected cells are reported to encode viral DUB, and these vial DUBs actively disrupt cellular Ub-dependent processes to suppress host antiviral immune response, enhancing virus replication and thus proliferation. This review surveys the types of DUBs encoded by different viruses and their molecular processes for how the infecting viruses take advantage of the DUB system to evade the host immune response and expedite their replication.

## 1. Introduction

Post-translational modifications of proteins by ubiquitin (Ub) and Ub-like modifiers (Ubl), such as Nedd8, interferon-stimulated gene 15 (ISG15), and small Ub-like modifier (SUMO), have emerged as major regulatory mechanisms in various aspects of cellular activities, including signal transduction, transcription, membrane protein trafficking, autophagy, nuclear transport, and immune responses [1,2,3,4,5]. Attachment of Ub and Ubl to target proteins is achieved by the sequential cascades of three enzymatic actions: those of an activating enzyme (E1), conjugating enzymes (E2), and ligases (E3) that determine target molecules with specificity [2,6]. These sequential enzymatic actions promote the degradation of a protein by the Ub–proteasome system (UPS), the most critical machinery for degradation of cytoplasmic and nuclear proteins [5,7].

Deubiquitination is a counteracting process to ubiquitination and is catalyzed by deubiquitinating enzymes, deubiquitinases (DUBs), which remove Ub from proteins by cleaving the peptide or isopeptide bond between Ub and its substrate proteins as well as between Ub molecules in a chain to reverse the fate of the ubiquitinated proteins and thereby the above noted molecular and cellular functions of the proteins [8,9,10,11,12]. The human genome encodes nearly 100 DUBs, which are classified into two main groups: cysteine proteases and metalloproteases. Cysteine protease DUBs consist of four major subfamily groups: (1) the Ub-specific protease (USP/UBP), the largest superfamily (various USPs, such as USP1, USP2, and so on) of DUBs; (2) the ovarian tumor (OTU) superfamily; (3) the Machado–Josephin domain (MJD) superfamily; and (4) the Ub C-terminal hydrolase (UCH) superfamily [13,14]. Meanwhile, metalloproteases include Jab1/Mov34/Mpr1 Pad1 N-terminal (MPN+) (JAMM) domain superfamily proteins, which bind zinc and are thus metalloproteases [14]. These DUBs contain a catalytic domain surrounded by one or more additional domains, such as Ub-specific proteases domain, Ubl domain, meprin and tumor necrosis factor (TNF) receptor associated factors (TRAF) homology (MATH) domain, zinc-finger Ub-specific protease (ZnF-UBP), etc. [13,15]. Conserved amino acid sequence motifs surrounding catalytically active amino acid residues and protein structures of these classes of DUBs are thoroughly reviewed in terms of mechanisms and interactive specificity [14].

Eukaryotic viruses take advantage of post-translational modifications for degradation of various viral and cellular proteins to overcome host defense mechanisms at different stages of the infection cycle. In these processes, both DNA and RNA viruses employ the cellular UPS for degradation [16,17,18,19], while many viruses also express their own E3s for degradation of cellular proteins, such as p53, major histocompatibility complex class I (MHC-I), etc. [20,21]. Hence, it is well established that viruses and their infected hosts exploit the proteasomal degradation system to promote efficient virus replication or to restrict invading viruses in the infected host, respectively. An overview of how ubiquitination and host and viral DUBs regulate protein fate is illustrated in Figure 1.

Viruses and the virus-infected cells also exploit DUBs to reverse biological actions of the ubiquitinated proteins by removing Ub from the target proteins throughout virus infection of the host, as reviewed [1,2]. However, differently from ubiquitination, not many examples for the targeting of protein deubiquitination by viruses have been described to date, leaving investigators without a comparably detailed molecular understanding of either the viral usurpation of deubiquitination or the responsive host infection blockade. This review is focused on how the infecting viruses and the invaded cells take advantage of deubiquitination systems for the competing biological objectives of virus proliferation and host cell preservation.

## 2. Cellular DUBs in Regulation of Virus Life Cycles

Upon virus infection, the invaded host activates antiviral innate immune signaling pathways to thwart virus replication [22]. The activation of the innate immune responses is initiated by recognition of pathogen-associated molecular patterns (PAMPs; the conserved components of microbes) via a large repertoire of pattern-recognition receptors (PRRs) [23,24]. The most well characterized PRRs in sensing different types of PAMPs are toll-like receptors (TLRs), retinoic acid-inducible gene I (RIG-I)-like receptors (RLR), NOD-like receptors, and C-type lectin receptors (CLRs) [22]. Recognition of PAMPs by specific PRRs activates their specific subsequent signaling pathways, inducing synthesis of various anti-viral molecules, such as interferons (IFNs) and pro-inflammatory cytokines [25,26,27,28]. During these processes, the host DUBs play a pivotal role in regulating the antiviral innate immune signaling pathways. For instance, USP7, a host DUB, prevents degradation of nuclear factor kappa-light-chain-enhancer of activated B cells (NF-κB) by deubiquitination from K48 of NF-κB p65 subunit in the TLR (TLR3 and TLR9) pathway to enhance stability of NF-kB, which upregulates the transcription of pro-inflammatory cytokines [29]. USP7-mediated enhancement of NF-κB activity is then counterbalanced by another DUB, known as A20, in the TLR pathway, wherein A20 down-regulates NF-κB activity by deubiquitinating TRAF6 [29,30]. In immune signaling cascade, DUBs, such as USP3 [31], USP15 [32], USP21 [33], cylindromatosis (CYLD) [34,35], etc., directly target the receptor RIG-I, a cytosolic RNA sensor, for K63-deubiquitination and type I IFN inhibition, while USP4 stabilizes RIG-I by removing the K-48 poly-Ub chain, positively regulating RIG-I-mediated anti-viral responses [36]. USP17 and USP25 also deubiquitinate RIG-I in a different manner; that is, USP17 stabilizes RIG-I by K48-deubiquitination, while USP25 inhibits RIG-I degradation by K63-deubiquitination [37,38].

Cellular DUBs, in addition to the regulation of the innate immune system, play an important role in alteration of the virus infectivity, replication, and pathogenicity. USP11 inhibits influenza virus infection [39], while USP14 augments replicability of a panel of viruses, including norovirus, encephalomyocarditis virus, Sindbis virus, and La Crosse virus [40]. USP7, which is also known as herpesvirus associated Ub-specific protease (HAUSP), is known to interact with ICP0 protein of herpes simplex virus type 1 (HSV-1), and the interactions between these two proteins are associated with viral infectivity [41]. Epstein-Barr nuclear antigen 1 (EBNA1) of Epstein-Barr virus (EBV) also utilizes USP7 to initiate disruption of PML nuclear bodies to contribute to nasopharyngeal carcinoma [42]. USP7 is also essential for EBNA1-mediated recruitment of a histone H2B deubiquitinating complex to the EBV latent origin to stimulate DNA binding activity of EBNA1 for the enhancement of virus replication [43]. Other reports further indicate that USP7 could interact with Kaposi’s sarcoma-associated herpesvirus (KSHV) latency-associated nuclear antigen (LANA) to regulate latent viral genome replication [44]. The Gaussia princeps luciferase protein complementation assay shows that USP15 interacts with the key oncoprotein human papillomavirus (HPV) E6, while USP29 and USP33 bind to E7 protein, suggesting that cellular DUBs impact HPV tumorigenesis by regulating the stability of the viral proteins [45]. USP15 is also known to be a critical determinant of the observed stability of HBx by direct interaction with HBx of HBV [46]. Retroviruses also take advantage of cellular DUBs for regulation of their life cycles, even if there have been no reports of the retrovirus-encoded DUBs to date. For instance, USP49 stabilizes APOBEC3G by deubiquitination and inhibits human immunodeficiency virus (HIV)-1 replication [47], while USP7 stabilizes HIV-1 trans-activator of transcription (Tat) and thereby increases HIV-1 production [48]. In the human T-cell leukemia virus type 1 (HTLV-1), the metalloprotease STAM-binding protein-like 1, a DUB, protects Tax oncoprotein from K48-induced ubiquitination to promote its shuttling from the nucleus to the cytoplasm, where it triggers NF-kB and NF-kB inhibitor kinase (IκK) activation [49]. Taken together, these reports indicate that cellular DUBs regulate virus replication and pathogenicity through different mechanisms in the infected host.

The infected viruses also express their own DUBs in the infected cells to circumvent the above addressed antiviral defense mechanisms imposed by the host, which is pivotal for the establishment of successful infection to generate replication competent progeny viruses. That is, the viruses take advantage of their encoded DUBs to reverse the function of E3 ligases involved in regulation of the reversible conjugation of Ub according to the types of virally expressed DUBs and their modes of action for immune evasion and virulence, as illustrated below. In a word, deubiquitination by cellular and viral DUBs is paramount to virus replication, which is governed by specifically accelerative or inhibitory DUBs.

## 3. Viral DUBs Essential for the Virus Life Cycles and Viral Defense

Upon virus infection, the infected viruses adopt strategies to avoid or delay activation of the host innate immune system, which plays an essential role in eliminating the infecting viruses [50,51]. Both DNA and RNA viruses exploit modification of Ub signaling by blocking activity or hijacking host machinery (E3 ligases or DUBs) to evade the immune response [51]. Further, the infected viruses encode their own E3 ligases or DUBs that can suppress the immune responses and/or dysregulate other Ub-dependent signaling pathways in the infected host [52,53,54]. Like host DUBs, viral DUBs also target multiple signaling molecules to overcome immune suppression imposed by the host. Our review here is focused on the virus-encoded deubiquitinases and their countering roles in suppressing the host’s innate immune activity, as listed in Table 1 and illustrated in Figure 2.

### 3.1. DUBs Encoded by the RNA Viruses

Not only DNA viruses but also RNA viruses encode DUBs to reverse Ub-mediated host cell processes [58,85,86,87,88], and some of these DUBs resemble OTU superfamily proteases [89,90,91,92,93,94] of the novel cysteine proteases [95].

The L (polymerase) segment of the tick-born Crimean-Congo hemorrhagic fever virus (CCHFV) [89,90,92] and Dugbe virus (DUGV), belonging to the genus Nairovirus of Bunyaviridae family [93], contains not only an RNA-dependent RNA polymerase but also a viral OTU domain protease homologue. The OTU domains in the L polymerase of both CCHFV and DUGV show DUB and ISGylase activities, actively processing Lys48 and Lys63 poly-Ub chains and ISG15 on cellular proteins [55], just as the OTU domain of A20 negatively regulates NF-κB signaling via deubiquitination [30,96]. Expression of OTU domain further reduced NF-kB-mediated promoter activity during TNFα stimulation and RIG-I-mediated IFN-β expression, indicating that the enzyme inhibits both antiviral and pro-inflammatory actions of the innate immune responses by the infection [56].

Equine arteritis virus (EAV) and porcine reproductive and respiratory syndrome virus (PRRSV), a member of the Arteriviridae family, contain novel OTU family of DUBs in the nonstructural protein 2 (nsp2) protease, and this domain has been characterized before as a papain-like protease domain 2 (PLP2) that is involved in the processing of replicase polyprotein [56]. Both in vitro and cell-based assays indicated that PLP2 DUB activity is conserved in all members of Arterivirus family, and ectopic expression of the PLP2 or full-length nsp2 of EAV lowered global levels of both Ub and ISG15 conjugates [55]. These results indicate that PLP2 has dual activities—viral protease to process replicase polyprotein as well as DUB. Expression of EAV PLP2 also suppresses activation of NF-κB by TNFα [55], suggesting that DUB activity is important for suppression of innate immune responses. The PLP2 impairs innate immunity by inhibiting RIG-I-mediated innate immune signaling by deubiquitination of RIG-I, when overexpressed [56], as is shown in the above Nairovirus DUBs. Likewise, PLP2 of PRRSV possesses DUB activity toward Ub and ISG15 conjugates [55,97], which inhibits NF-κB activity by inhibiting the Lys48-linked polyubiquitination of NF-κB inhibitor, alpha (IκBα) [57] and is responsible for evading RIG-I-mediated innate immune responses [56]. These results indicate that both Arteri- and Nairo-viruses employ their deubiquitinating potential to inactivate cellular proteins involved in RLR-mediated innate immune signaling [56].

The foot-and-mouth disease virus (FMDV), belonging to the Picornaviridae family harboring small non-enveloped positive sense ssRNA, causes disease by forming vesicles on the feet and mouths of the infected animals, thereby imposing substantial economic losses in the livestock industry. Infection of FMDV into a host produces a big polyprotein, and autoproteolytic cleavage of the nascent polyprotein generates the leader protease (L^pro^) from the N-terminus [98]. Since translation of this viral message is initiated by two in-frame initiation codons, two isoforms of L^pro^—Lab^pro^ and Lb^pro^—are synthesized, and the former has an additional 28 amino acid sequence at the N-terminus [98,99]. L^pro^ functions as a viral protease that cleaves not only viral proteins but also eukaryotic translation initiation factor 4 gamma (eIF4G). As eIF4G is critical for the translation initiation of the capped host messages, L^pro^-mediated cleavage impairs translation of the host mRNA, prioritizing translation of FMDV messages [100], which contain viral protein genome-linked (VPg) instead of the cap structure typifying the 5′ termini of most eukaryotic messages [101,102]. Mutational analyses indicated that L^pro^ plays a critical role in counteracting the cellular innate immune responses by inhibition of cellular IFN-α/β production and associated IFN-stimulated genes (ISGs), specifically double-stranded RNA-dependent protein kinase R, a cytoplasmic sensor of viral RNA [103,104,105,106,107]. Further mutational analysis showed that the conserved domain in the L^pro^ is required for proper subcellular localization, replication of FMDV, and expression of IFN-β [108]. The L^pro^ contains the PLP amino acid sequence domain, which shares homology with mouse hepatitis virus (MHV) PLP [109,110]. Further, the secondary structure of Lb^pro^ shows homology with severe acute respiratory syndrome coronavirus (SARS-CoV) PL^pro^ and USP14 [58], predicting that the protein may function as a DUB. In fact, ectopic expression of L^pro^ negatively regulated the interferon pathway by acting as a viral DUB [58]; antagonism of IFN and cleavage of eIF4G were uncoupled [58,108]. DUB function of L^pro^ was further demonstrated by over-expression of L^pro^, which reduced the amount of Ub-conjugated RIG-I, TANK binding kinase 1 (TBK1), TRAF3, and TRAF6 by deubiquitination, suppressing host Ub-dependent innate immune signaling pathways [58]. Mutations at the domains targeting SAF-A/B, Acinus, and PIAS (SAO) in L^pro^ abrogated DUB activity [58,108], again indicating that L^pro^ functions as a viral DUB. L^pro^ is also believed to play an important role in degradation of NF-κB [107]. Genomic sequence analysis of enterovirus G (EGV), also a member of Picornaviridae, shows that a nsp of EGV at the 2C/3A cleavage junction has amino acid sequence homology to torovirus PLP domain, which was subsequently named ToV-PLP, suggesting that the domain contains DUB and deISGylating activity. This suggestion was confirmed by experiments in which EGV lacking ToV-PLP domain increased cellular type I and II IFN production and ISG15 transcription [59], as FMDV does. These data prove the significance of viral DUBs in suppressing the innate immune response in the infected hosts.

Coronaviruses (CoVs) are enveloped viruses with positive sense ssRNA genomes (26~34 kb). Full-length genomes of the clinical isolates of SARS-CoV-1, belonging to the Coronaviruses family, were sequenced shortly after emergence of the initial SARS-CoV-1 pandemic in 2003 [111,112,113]. The sequence analysis showed that SARS-CoV-1 encodes PLP (PL^pro^) from nsp3 [114], and this PL^pro^ shows structural similarity to the PLP2 domain of cellular DUBs [115]. Further studies confirmed that this domain of SARS-CoV-1 indeed harbors both a protease and DUB activity [116,117,118], targeting both Ub and ISG15 to antagonize the induction of type I IFN to inhibit innate immunity [87,116,117,119,120]. Specifically, PL^pro^ inhibited RIG-I-, melanoma differentiation-associated protein 5 (MDA5)-, and TLR3-mediated IFN-β promoter activity and impeded signaling components specific to interleukin regulatory factor 3 (IRF3) activation to evade the host antiviral response [86,120,121]. It was later demonstrated that SARS-CoV-1 PL^pro^ decreased ubiquitinated forms of RIG-1, stimulator of interferon genes (STING), TRAF3, TBK1, and IRF3 to inhibit type I IFN signaling via the STING-TRAF3-TBK1 complex [60]. These results indicate that coronavirus PLP2 comprises multifunctional domains, a viral protease, DUB, and IFN antagonist, suggesting that each domain can serve as an independent target for antiviral therapeutics.

SARS-CoV-2, the etiologic agent of the current pandemic respiratory disease known as coronavirus disease 2019 (COVID-19), shares high genomic sequence identity with SARS-CoV-1 [122,123]. The virus expresses its own DUB, PL^pro^ (the protease domain of nsp3), which shows 83% sequence identity with SARS-CoV-1 PL^pro^ [124,125]. However, SARS-CoV-2 PL^pro^ has different preferences for the host substrate than SARS-CoV-1. Specifically, PL^pro^ of SARS-CoV-1 preferentially targets Ub chains, while the corresponding PL^pro^ of SARS-CoV-2 cleaves the Ub-like ISG15 [61]; i.e., SARS-CoV2 PL^pro^ cleaves ISG15 from IRF3 and thus attenuates type I IFN responses [61]. Further, inhibition of SARS-CoV-2 PL^pro^ activity with GRL-0617 impaired the virus-induced cytopathogenic effect as well as virus replication, while maintaining antiviral IFN, suggesting that SARS-CoV-2 PL^pro^ can be a dual target for suppressing SARS-CoV-2 infection and promoting antiviral immunity [61].

Another member of the coronavirus family, MERS-CoV (Middle East respiratory syndrome coronavirus) also encodes PL^pro^ with DUB activity. Structural analysis of MERS-CoV PL^pro^ revealed that the protease adopted a similar fold to SARS-CoV-1 PL^pro^, and the active site consisted of a Cys-His-Asp catalytic triad, even though the oxyanion hole appeared deficient in MERS-CoV PL^pro^ [126,127]. These reports indicate that CoV PL^pro^ domain contains two distinct functions—DUB to regulate protein stability and protease activity to process the viral polyprotein, which is controlled by the same active site. Mutational analyses show that abrogation of DUB activity by specific mutation did not disrupt cleavage of the viral polyprotein, indicating that these two enzymatic activities are separable [62]. Specifically, mitochondrial antiviral-signaling protein (MAVS)- and IRF3-mediated IFN-β activity was inhibited by PL^pro^ but not by DUB-deficient PL^pro^, intimating that DUB activity of PL^pro^ is directly involved in the evasion of cellular signaling pathways of innate immunity [62,63,64,65].

The MHV A59, a phylogenetic relative of SARS coronavirus, also expresses nsp3, which carries a conserved DUB motif within its PL^pro^ [117]. The PLP2 domain of PL^pro^ dysregulated RIG-I-, MAVS-, TBK1-, and IRF3-mediated signaling cascades [67,128] and thus inhibited cellular transcription of IFN-β [88]. Further mechanistic studies showed that the PLP2 catalytic domain of nsp3 of MHV A59 binds to IRF3, which causes deubiquitination and nuclear translocation of IRF3 and eventually reduces IRF3-mediated IFN-β induction [66,128,129]. In addition, porcine epidemic diarrhea virus (PEDV) and transmissible gastroenteritis virus (TGEV), enteric coronaviruses, also encode two PLP domains, called PLP1/PLP2 and PL1^pro^/PL2^pro^ [68,130], respectively, while avian infectious bronchitis virus (IBV), another member of the coronavirus family, expresses a single PLP (PL^pro^) [131]. These proteases function as DUBs, which also plays an important role in inhibition of IFN-β expression via RIG-I- and STING-triggered signaling pathways [68,130,131,132].

These reports collectively indicate that coronaviruses express viral proteins with DUB activity to reduce IFN-β, which contributes to rapid viral growth and thus SARS-associated immunopathology. These data also reveal that DUB and protease activities can be selectively incapacitated, opening a possibility of therapeutic development targeting the specific active site in PL^pro^.

Like these mammalian viruses, once the tymovirus turnip yellow mosaic virus (TYMV), a plant virus comprising positive sense ssRNA in its virion, infects its host, two nsps, p206 and p69, are expressed from two overlapping open reading frames (ORFs)—ORF-206 and -69, respectively [133]. The ORF-206 encodes the TYMV PLP (PRO) domain, which functions as Ub hydrolases (DUBs) [134,135,136], interfering with cellular processes critical to the virus replication. Briefly, upon TYMV infection, the host induces degradation of the viral RNA-dependent-RNA polymerase (RdRp) by tagging it with Lys48-linked poly-Ub and thereby driving it to the UPS [69], which impairs virus replication. However, TYMV PRO counteracts this cellular defense mechanism by removing the Lys-48-Ub chains from POL and thus stabilizes POL in a PRO-dependent manner, which activates virus replication [85,137].

Taken together, these reports demonstrate that diverse species of viruses, regardless of the polarity (positive- and negative-sense RNA viruses) and host (plant and mammalian viruses) express DUBs as a common strategy to overcome the hosts’ restrictions.

### 3.2. DUBs Expressed from the DNA Viruses

Like the RNA viruses, DNA viruses also encode specific DUBs to overcome restrictions imposed by infected cells. It is observed that antigen presentation of intracellularly synthesized viral proteins via cellular MHC-I is impeded during viral infection. Balakirev and coworkers hypothesized that impediment of antigen presentation in virus infected cells could be due to viral DUB-mediated reversion of ubiquitination processes necessary for the proteasomal degradation of proteins [70]. The hypothesis was confirmed by the finding that adenovirus (AdV), a non-enveloped, double-stranded DNA virus, encodes cysteine protease (Avp), which has structural similarity to the archetypal cysteine protease papain and functions as an AdV DUB by cleaving Lys-48-linked tetra Ub chains and ISG15 [70]. That is, Avp is essential for the proteolytic cleavage of the capsid protein of AdV for virus assembly and is also critical for the regulation of ISG15, which has antiviral activity through interfering with processes such as viral replication or translation [138]. These demonstrations provide the first example of a viral protease with deubiquitinating activity [70]. DUB activity of Avp was further supported by the consensus sequence of Avp with the C-terminal LRGG motif of Ub [139] and a significant reduction in the amount of ubiquitinated histone, H2A, in AdV-infected Hela cells, suggesting a potential ubiquitinated cellular target for Avp [140].

HSV-1, which causes cold sores, expresses Ub-reactive UL36^USP^ and is active toward both Lys48- and Lys63-linked poly-Ub chains, with a preference for Lys63 linkages [141,142]. Blocking the expression of UL36^USP^ in infected cells increased production of IFN-β, while virus containing wild-type UL36^USP^ impaired cellular IFN-β production, demonstrating that UL36^USP^ is an IFN antagonist [71]. Further molecular studies indicate that reductions in IFN-β promoter activity by UL36^USP^ was mediated by deubiquitination of TRAF3 [71] and decreased levels of Ub-conjugated IκBα, resulting in the inhibition of the NF-κB signaling cascade [72]. UL36^USP^ also inhibits the STING pathway, involved in detection of cytoplasmic viral DNA, and thus lowers cGAS/STING-mediated activation of IFN-β and NK-κB-responsive promoter [72]. This inhibition in turn lowers the production of IFN-β and IL-6 messages, presumably through deubiquitination of IκBα [72]. Other UL36 homologs with deubiquitinating activity, such as M48, UL48, and BPLF1 in murine cytomegalovirus (MCMV/β subfamily), human cytomegalovirus (HCMV/β subfamily) and EBV (γ subfamily), respectively, [143,144] are described below.

MCMV encodes Ub-specific cysteine protease, M48^USP^, a homolog of UL36^USP^, with DUB activity. Homology between M48^USP^ and UL36^USP^ is only 10%, but the putative catalytic triad Cys-His-Asp and an oxyanion hole-forming Gln are strictly conserved [143]. According to the crystal structure, M48^USP^ adopts a papain-like fold, wherein the active site cysteine forms a thioether linkage to the suicide substrate, and the exposed β hairpin loop of M48^USP^ is linked to the Ub core through hydrophobic interactions [145]. Incapacitation of the DUB activity of M48^USP^ significantly attenuated MCMV replication in mice by increases in MCK2, an MCMV-encoded pro-inflammatory chemokine, indicating that M48^USP^ plays an important role for the productive replication of the infected virus by regulating inflammatory processes [73].

EBV, another member of the gamma-herpesvirus subfamily, expresses N-terminus of viral tegument protein, BPLF1, with DUB activity, which interacts with EBV ribonucleotide reductase (RR) [146]. EBV RR consists of two subunits, a large (RR1) and small (RR2). Immunoprecipitation analysis indicates that BPLF1 interacts with RR2 [147]. Ubiquitination of RR1 requires expression of both subunits in the complex. Co-expression of BPLF1 with RR1 and RR2 eliminates ubiquitination of RR1, which is critical for inhibition of RR activity in a protease-dependent manner [147]. Taken together, these data show that EBV BPLF1 modulates the activity of EBV RR by deubiquitination. In addition to deubiquitination, BPLF1 is also involved in de-neddylation from Cullin, a scaffold protein involved in the formation of Cullin-RING Ub ligases (CRLs) [148], which is integral for permissive to EBV replication [148,149,150]. It has also been reported that BPLF1 promotes viral replication by reducing ubiquitination of proliferating cell nuclear antigen (PCNA) [151] and TRAF6 [74].

KSHV/human herpesvirus 8 (HHV-8), a member of gamma-herpesvirus family that encompasses EBV, murine gamma-herpesvirus 68 (MHV68), and herpesvirus Saimiri (HVS), encodes DUB ORF64 [75]. ORF64 reduces ubiquitination of RIG-I, which is ubiquitinated by tripartite motif protein 25 (TRIM25) [25,77,152,153] upon recognition of viral RNA to initiate IFN signaling cascade. Thus, ORF64 suppresses RIG-I-mediated signaling [75]. These ORF64-associated changes are known to be critical for the persistence of KSHV. Taken together, these reports indicate that DUB activity of ORF64 in KSHV is important for the counteraction of RIG-I, which senses KSHV infection.

Human cytomegalovirus (HCMV), MHV68, and Marek’s disease virus (MDV) also express corresponding homologs of UL36^USP^ with confirmed deubiquitinating activity [78,144]. Mutational analyses of the active DUB domains shed light on their biological roles. For instance, mutation in UL48 gene encoding the HCMV DUB (UL48^USP^) showed that UL48^USP^ played an essential role in oncogenesis by regulating PRR-mediated type I IFN via deubiquitination of TRAF3, TRAF6, interleukin-1 receptor-associated kinase 1 (IRAK1), IRF7, and STING signaling pathways [76]. Further, mutation in UL48^USP^ also reduced the production of infectious viral progeny during infection [144]. Likewise, inactivation of MDV-UL36^USP^ indicated that the domain shows linkage-specific DUB activity [79] and is key to the pathogenicity and maintenance of cellular transformation [79,80,154,155], even if specific cellular targets of the DUB were unclear. All of these reports collectively indicate that deubiquitination by viral DUBs in infected cells is vital to regulate oncogenesis as well as the virus life cycle.

Hepatitis B virus (HBV), a member of the Hepadnaviridae family, is a partially double strand relaxed circular DNA virus, causing hepatitis B [156]. Infection of HBV into the host cell expresses HBx oncoprotein, a cysteine protease. Expressed HBx inhibits type I IFN-mediated antiviral innate immune responses by removing K63-Ub from RIG-I via its N-terminal domain, again showing that innate immune receptor activity can be impeded by viral DUBs [81]. HBx also determines stability of PRR-associated proteins, such as STING, IRF3, TRAF3, etc., by ubiquitination and thereby production of IFN [82,83,84]. Conversely, the HBx expressed in infected cells can be actively ubiquitinated and is degraded by the proteasome pathway. It is reported that tumor suppressor protein p53 is capable of increasing HBx ubiquitination by an unknown mechanism [157]. Other cellular factors, such as 26 S proteasome complex [158], transcription factor Id-1 [159], MDM2 [157], Siah-1 [160], etc., also contribute to destabilization of HBx in a proteasome-dependent manner. Further study showed that USP15 was a critical determinant of the observed stability of HBx by direct interaction with HBx [46]. Taken together, these reports demonstrate that HBx is integral for suppression of the host immune response and is targeted for proteosomal degradation by various cellular factors.

## 4. Concluding Remarks and Future Directions

Ubiquitination is the process of attaching Ub to target proteins for proteasomal degradation, while deubiquitination antagonizes this process utilizing DUBs to cleave the peptide or isopeptide bond between Ub and its substrate proteins as well as between Ub molecules in a chain. Both ubiquitination and deubiquitination play a pivotal role in regulating the degradation of proteins via a proteasome and lysosome and thereby in regulating various cellular and molecular processes. Viruses also exploit DUBs as much as the UPS. The dynamic reciprocal counteractions between viruses and hosts using their specific DUBs are powerful strategies for their proliferation and restrictions.

It appears that encoding of viral DUBs is universal to the survival and proliferation of viruses in infected hosts, respecting that DUBs were found in diverse virus lineages, as noted above, regardless of RNA or DNA class, negative- or positive-sense RNA type, or animal or plant source. Even if it were reasonable to assume that all viruses might take advantage of the DUB system, based on these findings, not all DUBs have yet been identified in all viruses. More viral DUBs apart from the above illustrated viruses need to be identified and characterized to substantiate the possibility that all viruses employ DUBs to elude the hosts’ antiviral immune activities. Identification and characterization of viral DUBs used by additional viruses will help establish the universality and versatility of viral defense mechanism in infected hosts, which will greatly inform development of antiviral therapeutics targeting viral DUBs.

As reviewed here and elsewhere [51,140,161], interestingly, viral proteins with DUB activity are in most cases viral proteases that are vital for processing polyproteins into mature functional cleavage products. Cleavage of the polyprotein by the viral protease is especially critical for the positive sensed ssRNA viruses, since these viruses produce one massive polyprotein from a single open reading frame and thus mutation in the protease region cannot generate a mature functional viral protein. Functional knockout of the viral protease gene can completely incapacitate the viral action, suggesting that the viral protease will be an ideal target for antiviral therapeutics. Moreover, these viral proteases are multifunctional, such as in the processing of viral polyproteins, stability of antiviral immune response molecules, subcellular localization of cellular DUBs, and so on [162,163,164,165,166]. Thus, the structural and sequential analyses focusing on the specific domains (motifs) for individual functions are integral for revealing whether the viral protease activities are determined by a single active site or multiple distinct sites in regulating particular actions of the protease. These studies will provide important information on selective inhibition of multifunctional viral protease activities, pursuant to designing therapeutics against specific viruses.

Finally, viral DUBs encoded from different viruses, even if their catalytic triads are situated at disparate sites, could target identical molecules to inhibit antiviral pathways in the infected cells. This approach can be applied to cellular DUBs as well. For instance, USP14 alters replicability of multiple viruses, such as norovirus, encephalomyocarditis virus, Sindbils virus, and La Crosse virus [40]. These facts point to a common therapeutic principle based on viral and cellular DUBs designed to act on multiple respective cellular and viral DUBs in a pleiotropic manner for patients co-infected with multiple viruses.

## Figures and Tables

**Figure 1 ijms-22-04438-f001:**
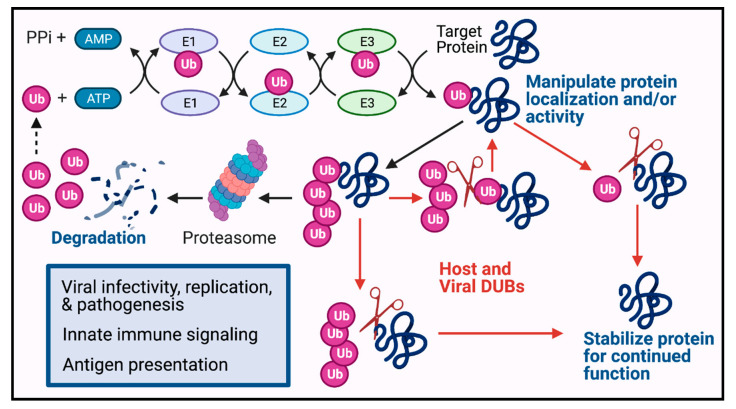
Deubiquitination counteracts ubiquitination within the UPS by cleaving the peptide or isopeptide bond between Ub and its substrate proteins as well as between Ub molecules in a chain to reverse the fate of the ubiquitinated proteins. Viruses utilize DUBs to evade host immune response and expedite their survival. DUB pathways are illustrated by red arrows. Dotted lines denote Ub recycling. Blue text indicates ‘protein fate’ which coordinates outcomes that are regulated by the Ub system during host–viral interactions (blue box). Image made with BioRender.com.

**Figure 2 ijms-22-04438-f002:**
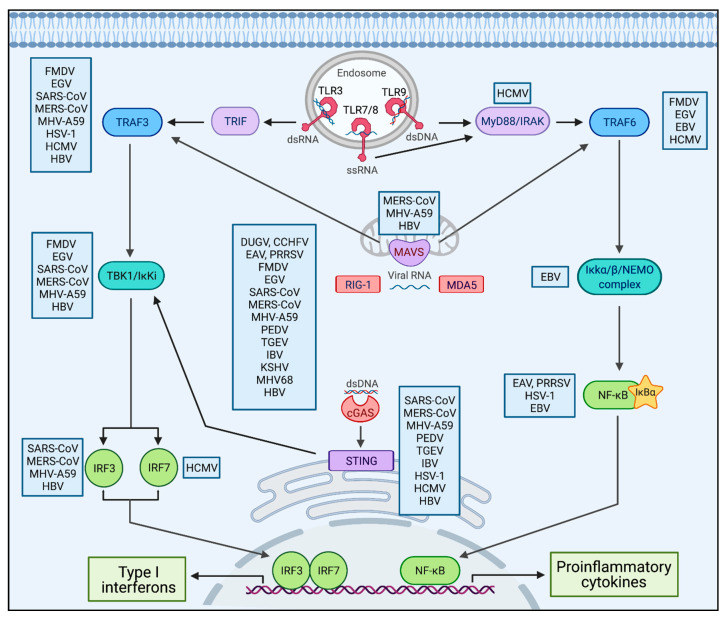
Viruses block innate immune signaling pathways using DUB activity. Viral recognition (red) initiates a cascade of signaling mediators (purple/blue/turquoise) leading to the activation of transcription factors (green) to induce expression of proinflammatory cytokines and type I interferons (green boxes). Viruses (blue boxes) exploit virally encoded DUBs to block different steps within innate immune signaling pathways. Image made with BioRender.com.

**Table 1 ijms-22-04438-t001:** Viral DUBs and their cellular targets. # and ? indicate viral target or unknown host target, respectively. Host targets in parentheses indicate potential targets based on homology among viral DUBs and/or identified regulation/interaction that has not yet been attributed to DUB activity. Abbv: DUGV—Dugbe virus; CCHFV—Crimean-Congo hemorrhagic fever virus; EAV—equine arteritis virus; PRRSV—porcine reproductive and respiratory syndrome virus; FMDV—foot-and-mouth disease virus; EGV—enterovirus G; SARS-CoV—severe acute respiratory syndrome coronavirus; MERS-CoV—Middle East respiratory syndrome coronavirus; MHV-A59—mouse hepatitis virus A59; PEDV—porcine epidemic diarrhea virus; TGEV—transmissible gastroenteritis virus; IBV—infectious bronchitis virus; TYMV—tymovirus turnip yellow mosaic virus; AdV—adenovirus; HSV-1—herpes simplex virus 1; MCMV—murine cytomegalovirus; EBV—Epstein-Barr virus; KSHV—Kaposi’s sarcoma-associated herpesvirus; HCMV—human cytomegalovirus; MHV68—murine gamma-herpesvirus 68; MDV—Marek’s disease virus; HBV—hepatitis B virus.

Genome	Virus	Viral DUBs	Host Targets	Cite
RNA	DUGV, CCHFV	L polymerase	RIG-I	[55,56]
	EAV, PRRSV	Nsp2	RIG-I and IκBα	[55,56,57]
	FMDV	L^pro^	RIG-I, TBK1, TRAF3, and TRAF6	[58]
	EGV	ToV-PLP domain at the 2C/3A junction	RIG-I, TBK1, TRAF3, and TRAF6	[59]
	SARS-CoV-1	Nsp3-PL^pro^	IRF3, RIG-I, TRAF3, TBK1, and STING	[60]
	SARS-CoV-2	Nsp3-PL^pro^	IRF3 (RIG-I, TRAF3, TBK1, STING)	[61]
	MERS-CoV	Nsp3-PL^pro^	MAVS and IRF3 (RIG-I, TRAF3, TBK1, STING)	[62,63,64,65]
	MHV-A59	Nsp3-PL^pro^	IRF3 and TBK1 (RIG-I, MAVS, TRAF3, STING)	[64]
	PEDV	PL^pro^	RIG-I, STING	[66]
	TGEV	PL^pro^	? (RIG-I, STING)	[67]
	IBV	PL^pro^	? (RIG-I, STING)	[68]
	TYMV	ORF-206	Viral RdRp ^#^	[69]
	AdV	Avp	H2A	[70]
DNA	HSV-1	UL36^USP^	TRAF3, IκBα, and STING	[71,72]
	MCMV	M48^USP^	?	[73]
	EBV	BPLF1	TRAF6, NEMO, and IκBα	[74]
	KSHV	ORF64	RIG-I	[75]
	HCMV	UL48^USP^	TRAF3, TRAF6, IRAK1, IRF7, and STING	[76]
	MHV68	ORF64	? (RIG-1)	[77]
	MDV	MDV UL36^USP^	?	[76,78,79,80]
Relaxed circular DNA	HBV	HBx	RIG-I, IRF3, TRAF3, IκKi, MAVS, and STING	[81,82,83,84]

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
