# Peer review of "Role of Virally-Encoded Deubiquitinating Enzymes in Regulation of the Virus Life Cycle"

_ijms, 2021, doi:10.3390/ijms22094438_

Round 1

Reviewer 1 Report

Proulx et al. in their review entitled “Role of deubiquitinating enzymes in regulation of the virus life cycle’ provide a straight-forward, comprehensive summary of the types of DUBs encoded by different viruses and how the viruses use the DUB system to evade host immune responses within the viral life cycle. Although there are other reviews covering DUBs in the viral life cycle, this review focused on virally-encoded DUBs and therefore addressed an area of need within the field. Minor comments/suggestions are provided below for the authors’ consideration:

  1. Since this review is focused on the virus-encoded deubiquitinases and their roles in suppressing the host’s innate immune activity, the title would be more descriptive if it was changed to: ‘Role of virally-encoded deubiquitinating enzymes in regulation of the virus life cycle’.
  2. Are there any examples of retroviral (human or animal)-encoded deubiquitinases?  It is interesting that various RNA and DNA viruses encode DUBs, but retroviruses seem to be the exception. There are several examples of cellular DUBs which regulate retrovirus lifecycles and could be included in ‘Section 2. Cellular DUBs in regulation of virus life cycles’. This would provide a more comprehensive overview of the various types of viruses (DNA, RNA, relaxed circular DNA, retroviral) regulated by cellular DUBs.

Author Response

Reviewer #1:

  • Since this review is focused on the virus-encoded deubiquitinases and their roles in suppressing the host’s innate immune activity, the title would be more descriptive if it was changed to: ‘Role of virally-encoded deubiquitinating enzymes in regulation of the virus life cycle’.

[Response] We have amended the title accordingly. Page 1 line 2.

  • Are there any examples of retroviral (human or animal)-encoded deubiquitinases? It is interesting that various RNA and DNA viruses encode DUBs, but retroviruses seem to be the exception. There are several examples of cellular DUBs which regulate retrovirus life cycles and could be included in ‘Section 2. Cellular DUBs in regulation of virus life cycles’. This would provide a more comprehensive overview of the various types of viruses (DNA, RNA, relaxed circular DNA, retroviral) regulated by cellular DUBs. 

[Response] The recommended inclusion of any possible retroviral DUBs, for completeness, would be desirable, but the follow-up database queries on these themes have only retrieved cellular DUBs major to innate immunity regulation.  Thus, we have updated Section 2 (pg 3 lines 119-127) to give proportionate attention to retroviruses in regard to cellular-DUB regulation of retrovirus life cycles.

Reviewer 2 Report

This review article by Proulx and others nicely summarizes the published papers on the relationships between ubiquitin system and viruses. This reviewer suggest some points that will further improve the manuscript. 

This reviewer found many inappropriate citations. For example, ref [42-45] (page 3 line 107) must be citations on KSHV, but only #44 appears to deal with KSHV. Are the following citations really the ones they intend: ref [47] (page 3 line 121), ref [122] (line 304), ref [123] (line 307)? 

page 3 line 99-113: although the authors recap that ".. cellular DUBs restrict virus replication and pathogenesis. (line 120-121)", it seems to this reviewer that the examples in this paragraph are exploitations of cellular DUBs by viruses.

page 3 line 114-121: this paragraph describes viral DUBs, but the last sentence says "cellular DUBs" (line 120). In addition, the last sentence of the previous paragraph (line 111-113) seems contradictory to the last sentence of this paragraph (line 120-121). This reviewer understand that the both sentences can hold true, but writing of this part (line 94-121) is not very logical. 

Fig1: this figure is too general and not informative. Every reader knows about this simple scheme even without seeing it. The authors need to be more specific. For example, since Table 1 shows many viral factors that affect host innate immunity, how about drawing a picture of innate immunity signals, in which some viral DUBs are included. 

Author Response

Reviewer #2:

  • This reviewer found many inappropriate citations. For example, ref [42-45] (page 3 line 107) must be citations on KSHV, but only #44appears to deal with KSHV.  Are the following citations really the ones they intend: ref [47] (page 3 line 121), ref [122] (line 304), ref[123] (line 307)? 

[Response] We appreciate this reviewer’s observations on the referencing and have made the needed corrections: ref [42-45] (page 3 line 107) now reads ref [44] (line 115); ref [47] (page 3 line 121) was removed (pg 4 line 138); ref [122] (line 304) now reads ref [124] (line 321), ref [123] (line 307) now reads ref [125] (line 324).

  • page 3 line 99-113: although the authors recap that ".. cellular DUBs restrict virus replication and pathogenesis. (line 120-121)", it seems to this reviewer that the examples in this paragraph are exploitations of cellular DUBs by viruses. 

[Response] We invoked cellular DUBs briefly to illustrate similarities between viral and cellular DUB structures, and between strategies exploited by the infected viruses and by the invaded hosts utilizing DUBs. We have revised the statement: “cellular DUBs restrict virus replication and pathogenicity” to now read: “cellular DUBs regulate virus replication and pathogenicity”. Page 3 line 128.

  • page 3 line 114-121: this paragraph describes viral DUBs, but the last sentence says "cellular DUBs" (line 120). In addition, the last sentence of the previous paragraph (line 111-113) seems contradictory to the last sentence of this paragraph (line 120-121). This reviewer understand that the both sentences can hold true, but writing of this part (line 94-121) is not very logical. 

[Response] To address the reviewers concern regarding mention of cellular versus viral DUBs on page 3 line 114-121, we have revised the sentence to read: “deubiquitination by cellular and viral DUBs is paramount to virus replication” (pg 4 line 136-137). Moreover, in addressing the previous critique by revising the phrase ‘restrict’ to ‘regulate’ (pg 3 line 128) as well as the current critique, we feel these statements no longer read as contradictory.

  • Fig1: this figure is too general and not informative. Every reader knows about this simple scheme even without seeing it. The authors need to be more specific. For example, since Table 1 shows many viral factors that affect host innate immunity, how about drawing a picture of innate immunity signals, in which some viral DUBs are included.

[Response] We have decided to maintain figure 1 as a general overview of the ubiquitination/deubiquitination system to ensure simplicity and understanding to all readers. However, we agree that a more detailed figure illustrating some of the host-viral DUB interactions in the innate immune response (as shown in table 1) would enrich our manuscript. Thus, we have also added a figure 2 as a more detailed illustration for these interactions (pg 5 lines 161-165). We now reference Figure 2 alongside Table 1 in the text (pg 4 lines 149-150). Moreover, we moved Figure 1 to Section 1 (pg 2 lines 72-77) and reference Figure one in the text as “An overview of how ubiquitination and host and viral DUBs regulate protein fate” (pg 2 lines 60-61).

Round 2

Reviewer 2 Report

Fig2 looks good.